# A Corpus of Annotated Revisions for Studying Argumentative Writing

## Abstract

This paper presents a new corpus of annotated revisions of argumentative essays. This corpus analyzes between-draft revisions in the context of the full essay. The writer's intention for each revision is labeled with categories analogous to those used for argument mining and discourse analysis. The corpus enables more advanced research in writing comparison and revision analysis. Applications of the corpus are demonstrated via a study on student revision behaviors and a study on automatic revision intention prediction.

## 1 Introduction

Much of the writing-related NLP research focuses on the analysis of single drafts. Examples include document-level quality assessment (Attali and Burstein, 2006; Burstein and Chodorow, 1999), discourse-level analysis and mining (Brown and Yule, 1983; Burstein et al., 2003; Falakmasir et al., 2014; Persing and Ng, 2016), as well as fine-grained phrasal-level error detection (Leacock et al., 2010; Grammarly, 2016). Less studied is the analysis of *between drafts* – a comparison of revisions and the properties of the differences. Research on the topic allows a variety of applications: revision analysis (Zhang and Litman, 2015), paraphrase (Malakasiotis and Androutsopoulos, 2011) and correction detection (Swanson and Yamangil, 2012; Xue and Hwa, 2014).

Although there are some corpora resources for NLP research on writing comparisons, most tend to be between individual sentences/phrases for tasks such as paraphrase comparison (Dolan and Brockett, 2005; Tan and Lee, 2014) or grammar error correction (Dahlmeier et al., 2013; Yannakoudakis et al., 2011). In terms of revision anal-

ysis, the most relevant work are on Wikipedia revisions (Daxenberger and Gurevych, 2013; Bronner and Monz, 2012); however, the domain of Wikipedia is so specialized that the properties of their revisions do not correspond well with other kinds of texts.

This work presents an annotated corpus to facilitate revision analysis for argumentative essays. The corpus consists of a collection of three drafts of essays written by college students; the drafts are manually aligned at the sentence level, and the purpose of each revision is manually coded, using a revision schema closely related to argument mining/discourse analysis. Within the domain of argumentative essays, the corpus may be used for research and application of argument mining techniques and argumentative revision analysis. Outside of the domain, the corpus may also be of interest to research on paraphrase comparisons, grammar error correction, and computational stylistics (Popescu and Dinu, 2008; Flekova et al., 2016). We expect the corpus to be useful for advanced writing comparison study (discourse level) that connects to the mainstream research on writing analysis (e.g., argument mining). In this paper, we present two applications of our corpus: 1) Rewriting behavior data analysis 2) Automatic revision purpose classification.

## 2 Corpus Design Decisions

Consider this scenario: Alice begins her social science argumentative essay with the sentence: "Electronic communication allows people to make connections beyond physical limits."

An analytical system might (rightly) identify the sentence as the thesis of her essay, and an evaluative system might give the essay a lower score due to this sentence's vagueness and lack of evidence (though Alice may not know why she re-

ceived that score).

Now suppose in a revised draft, Alice expanded the sentence: "Electronic communication allows people to make connections beyond physical ~~limits~~ *location and enriches connections that would have been impossible to make otherwise.*"

An analytical system would still identify the sentence as the thesis, and an evaluative system might raise the overall score a little higher. Alice may become satisfied with the increase and move on. However, there is an opportunity lost – neither systems addressed the quality of her revision.

A revision analysis system might be helpful for Alice because it would link "limits" to "location and ..." and identify the reason why she made the change – perhaps *adding precision.* If Alice had intended her change as a way to add evidential support for her thesis, she would see that her attempt was not as successful as she hoped.

The above scenario highlights the application of a revision analysis system. This paper is about creating a corpus to enable the development of such systems. Because this is a relatively new problem, there are many possible ways for us to design the corpus. Here, we discuss some of our decisions.

First, we need to define the unit of revision. In the example above illustrates a phrase-aligned revision. While this offers a more precise definition of the scope of a revision, it may be difficult to achieve consistent annotations. For example, the changes may not adhere to any syntactic linguistic unit. For this first corpus, we define our unit of revision to be at the sentence level. In other words, even if a pair of sentences contain multiple edits, the entire sentence pair will be annotated as one sentence revision.

Second, we need to define the quality we want to observe about the revision sentence pair. For this first corpus, we focus on recognizing the purpose of the revision, as in the example above. It is a useful property, and it has previously been studied by others in the literature. People have considered both binary purpose categories such as Content vs. Surface (Faigley and Witte, 1981) or Factual vs. Fluency (Bronner and Monz, 2012); and more fine-grained categories (Pfeil et al., 2006; Jones, 2008; Liu and Ram, 2009; Daxenberger and Gurevych, 2012; Zhang and Litman, 2015). Our corpus follows the two-tiered schema used by (Zhang and Litman, 2015) (see Section 3.2).

Third, we not only have to decide on the an-

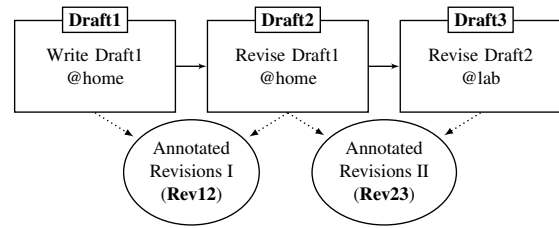

Figure 1: Our collected corpus contains five components: three drafts of an essay and two annotated revisions between drafts.

notation format, we also need to decide on how to obtain the raw text: argumentative essays with multiple drafts. We decided to sample from a population of predominantly college students, inclusive of both native and proficient non-native (aka L2) speakers. Comparing to high school students, college students are expected to have a better organization of the argument elements. Including native and L2 speakers allows for the exploration of possible rewriting differences between writers of varying backgrounds. We decided to give all subjects the same writing prompt and collect three drafts. The identical prompt minimizes the impact of topic difference for argumentation-related study. The collection of three drafts allows for a comparison of revision differences at different stages of rewriting.

Finally, we need a method of eliciting two revised drafts from each writer. Ideally, an instructor would give formative feedback after each draft for each student, but we do not have the resources to carry out such an expensive project. We decided to simulate instructor feedback by asking students to add more examples after the first draft. To elicit a second revised draft, we tried two strategies: 1) show students a character-based comparison between their first and second draft,[1]; and 2) show students what an idealized revision analysis system might tell them.

## 3 The Argumentative Revision Corpus

Based on the above design decisions, we have developed a corpus of argumentative essays with three drafts and detailed annotations for sentence-aligned revisions between each consecutive pair of drafts. The main corpus has five elements; the relationships between them are shown in Figure 1;

---

[1]Code derived from https://code.google.com/p/google-diff-match-patch/ which implements Myers' algorithm (Myers, 1986).

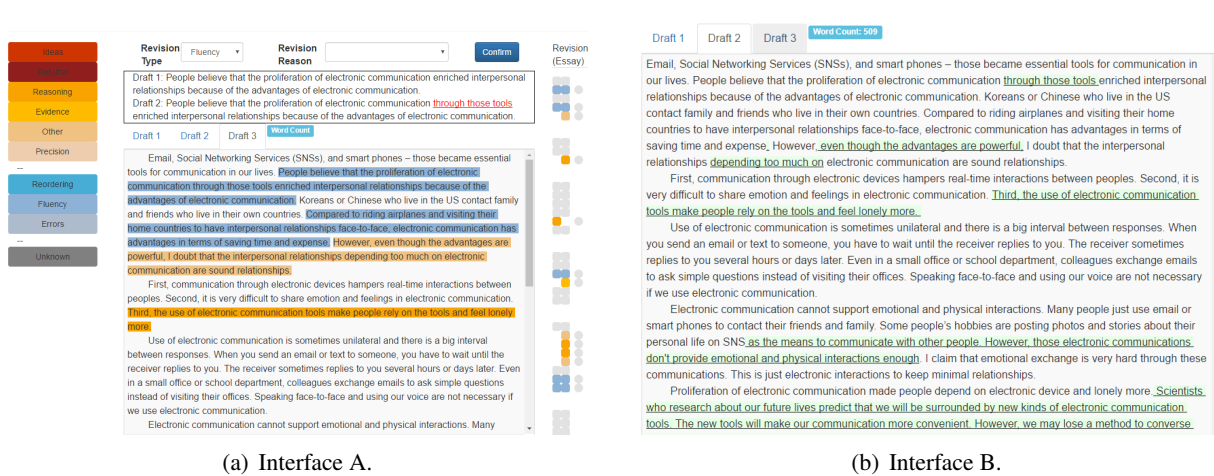

(a) *Interface A.*  (b) *Interface B.*

Figure 2: Screenshot of the interfaces. (a) *Interface A* with the annotated revision purposes, (b) *Interface B* with a streamlined character-based diff.

Section 3.1 describes the procedure of obtaining them. Section 3.2 briefly describes the revision schema we used, and reports the inter-annotator agreement. Additionally, we have collected metadata from the participants who contributed to the corpus (discussed in Section 3.3); these data may be useful for user behavior analysis.

### 3.1 Corpus Development Procedure

We have recruited 60 participants aged 18 years and older, among whom 40 were English native speakers and 20 were non-native speakers with sufficient English proficiency.[2] The study is carried out in three 40-60 minute sessions over the duration of two weeks.

**Draft1** Each participant begins by completing a pre-study questionnaire (Section 3.3) and writing a short essay online. Participants are instructed to keep the essay around 400 words, making a single main point with two supporting examples. They are given the following prompt:

> *"Suppose you've been asked to contribute a short op-ed piece for The New York Times. Argue whether the proliferation of electronic communications (e.g., email, text or other social media) enriches or hinders the development of interpersonal relationships."*

**Draft2** A few days after, participants are asked to revise their first draft online based on the following feedback: Strengthen the essay by adding one more example or reasoning for the claim; then

add a rebuttal to an opposing idea; keep the essay at 400 words.

**Annotated Revisions I (Rev12)** The two drafts are semi-manually aligned at the sentence level.[3] Then, the purpose of each pair of sentence revision is manually coded by a trained annotator, following the annotation guideline (see Section 3.2).

**Draft3** Participants perform their third revision in a lab environment. This time, they are not given additional instructional feedback. Instead, participants are shown a computer interface that highlights the differences between their first and second draft. They are asked to revise the third draft to improve the general quality of their essay. We experimented with two variations of elicitation. Chosen at random, half of the participants are shown Interface A, an interface that highlights the annotated differences between the drafts (Figure 2(a)); half of the participants are shown Interface B, a streamlined character-based diff (Figure 2(b)). Both groups are asked to read a tutorial about their respective interfaces before beginning to revise. Additionally, participants in group A are also asked to verify the manually annotated revision purposes between their first and second draft. After completing the final revision, all participants are given a post-study survey about their experiences (Section 3.3). Additionally, participants in group A are asked to verify the automatically predicted revision purposes between their second and

---

[2]i.e., with a TOEFL score higher than 100.

[3]Sentences are first automatically aligned using the algorithm in Zhang and Litman (2014) and then manually corrected by human.

| Draft1 | Revision Purpose | Draft2 | Revision Purpose | Draft3 |
|---|---|---|---|---|
| This world has no restriction on who one can talk to. | Conventions/ Grammar/ Spelling | This world has no restrictions on whom one can talk to. | | This world has no restrictions on whom one can talk to. |
| | | | Rebuttal/ Reserva-tion | Unfortunately, the younger users of digital communication cannot be entirely protected from the rhetoric of any out-sider. |
| | | | Warrant/ Reasoning/ Backing | Modern society is now faced with the issue of cyber bullying as a result. |
| The only aspects of communica-tion that this new development improves are internet navigation and faux internet relatability. | Word-Usage/ Clarity | The only aspects of digital com-munication that this new de-velopment improves are internet navigation and faux internet re-latability. | Word-Usage/ Clarity | The only aspects of digital com-munication that this new de-velopment improves are internet navigation and faux internet re-lationships. |
| | Claims/ Ideas | Being immersed in the sphere of new technologies can allow for complete isolation from the ac-tive, non-digital world. | | Being immersed in the sphere of new technologies can allow for complete isolation from the ac-tive, non-digital world. |

Table 1: An example of the annotated corpus. The sentences were aligned across the drafts and the revision purposes were labeled on the aligned sentence pairs. From Draft1 to Draft2, there are two *Modify* revisions (*Spelling* and *Clarity*) and one *Add* revision. From Draft2 to Draft3, there are two *Add* revisions (*Rebuttal* and *Reasoning*) and one *Modify* revision (*Clarity*)

third draft (Section 4.2).

**Annotated Revisions II (Rev23)** Regardless of which interface the participant used, the second and third draft are compared and annotated by the trained annotator in the same process as before.

### 3.2 Revision Annotation Guideline

Following prior codings, sentence revisions are first coarsely categorized as *surface* or *content-based* changes (Faigley and Witte, 1981), de-pending on whether any informational con-tent was modified; within each, we distin-guish between several finer categories based on the argumentative and discourse writing litera-ture (Kneupper, 1978; Faigley and Witte, 1981; Burstein et al., 2003). Our adapted schema has three *Surface* categories: Organization, Conven-tions/Grammar/Spelling, and Word-usage/Clarity; and five *Content-based* categories: Claim/Ideas, Warrant/Reasoning/Backing, Evidence, Rebut-tal/Reservation, and General Content Develop-ment. Table 1 shows examples of the aligned sen-tences in the three collected drafts and their anno-tated revision categories.

Two annotators (one is experienced, and the other is newly trained) annotated the files sepa-rately and discussed on the disagreed annotations to remove possible annotation errors due to misun-derstanding. After clearing the possible errors, the Kappa on the held-out data is 0.84 on two high-level categories of *Surface* vs. *Content-based*; and 0.71 on their eight low-level categories. The files labeled by the trained annotator were used as the gold standard annotation after the corrections.

### 3.3 Meta-Data

In addition to the raw text and the annotations, this corpus release also includes meta-data col-lected about the participants. This includes: a pre-study survey, a post-study survey, and demo-graphic statistics.

**Pre-Study Survey** The pre-study survey con-tains participant demographic information as well as their self-reported writing background, such as their confidence in their writing ability, the num-ber of drafts they typically make, etc.

**Post-Study Survey** The post-study survey con-tains questions about the participants' in-lab revi-sion experience, such as whether they found the computer interface helpful. All questions are an-swered on a scale of 5, ranging from "strongly dis-agree" to "strongly agree".

**Demographic Statistics** Among the 40 native speakers, there were 29 (72.5%) undergraduates, 6 (15%) graduate students, and 5 (12.5%) non-

students. Among the 20 L2 speakers, there were 4 (20%) undergraduates, and 16 (80%) graduate students; there were 9 Chinese, 2 Bengali, 2 Marathi, 2 Persian, 1 Arabic, 1 Korean, 1 Portuguese, 1 Spanish, and 1 Tamil. In terms of their disciplines, 33 participants (55.9%) were from the natural sciences, 24 (40.7%) from the social sciences, and 2 (3.4%) from the humanities.

### 3.4 Public Release

This corpus will be freely available for research purposes, with the first release coordinated with the publication of this paper. This version release will include: the raw text plus revision annotations, and the meta-data. The revision annotations are stored as .xlsx files. There are 60 spreadsheet files for revisions from Draft1 to Draft2 and 60 more spreadsheet files for revisions from Draft2 to Draft3. Each spreadsheet file contains two sheets: Old Draft and New Draft. Each row in the sheet represents one sentence in the corresponding draft. The index of the aligned sentence row in the other draft and the type of the revision on the sentence are recorded. The meta-data are in .log text files. Information in the text files are stored as the JSON data format.

## 4 Applications of the Corpus

While the development of a full fledged revision analysis system is outside the scope of this work, we demonstrate the potential applications of our corpus with two examples. The first performs statistical analyses on the collected revision data and meta-data to understand aspects of participant behaviors. The second uses the corpus to train a supervised classifier to automatically predict revision purposes.

### 4.1 Application: Student Revision Behavior Analysis

While it is well-established that thoughtful revisions improve one's writing, and while many college-level writing courses require students to submit multiple drafts on their writing assignments (Addison and McGee, 2010), instructors rarely monitor and provide feedback to students while they revise, partly due to time constraints, partly due to their uncertainty about how to support students revisions (Cole, 2014; Melzer, 2014). There is much we do not know about how to stimulate students to self-reflect and revise.

|  | Content | | Surface | |
|---|---|---|---|---|
|  | Rev12 | Rev23 | Rev12 | Rev23 |
| L2 (20) | 172 | 78 | 163 | 176 |
| Interface A | 91 | 37 | 71 | 85 |
| Interface B | 81 | 41 | 92 | 91 |
| Native (40) | 334 | 285 | 303 | 246 |
| Interface A | 177 | 154 | 149 | 111 |
| Interface B | 157 | 131 | 154 | 135 |

Table 2: Number of revisions, by participant groups (language, interface), coarse-grain purposes, and revision drafts (Rev12 is between Draft1-Draft2; Rev23 is between Draft2-Draft3.

#### 4.1.1 Hypotheses

Using the Argumentative Revision Corpus, we can begin to ask and address some questions about student revision habits and behaviors. Our first question is: How do different types of revision feedback impact student revision? And relatedly: Does student background (native vs. L2) make a difference?

We mine the corpus to test the following hypotheses:

**H1.** There is a difference in participants' revising behaviors depending on which interface is used to elicit the third draft.

**H2.** For participants who used Interface A, if the recognized revision purpose differs from the participants' revision intention, participants will further modify their revision.

**H3.** L2 and native speakers have different behaviors in making revisions.

**H1** and **H2** address the first question; **H3** addresses the second.

#### 4.1.2 Methodology

To test the hypotheses, we will use both subjective and objective measures. Subjective measures are based on participant post-study survey answers. Ideally, objective measures should be based on an assessment of improvements in the revised drafts; since we do not have evaluative data at this time, we approximate the degree of improvement with the number of revisions, since these two quantities were demonstrated to be positively correlated (Zhang and Litman, 2015). The objective measures are computed from Tables 2 and 3.

To compare differences between specific subgroups on the subjective and objective measures, we conduct ANOVA tests with two factors. For example, one factor is the native language of the participant, and another is the interface used. To

| Revision Purpose | Draft1 to Draft2 | | | Draft2 to Draft3 | | |
|---|---|---|---|---|---|---|
| | #Add | #Delete | #Modify | #Add | #Delete | #Modify |
| *Content* | 294 | 179 | 33 | 320 | 27 | 16 |
| Claims/Ideas | 25 | 8 | 4 | 5 | 0 | 0 |
| Warrant/Reasoning/Backing | 166 | 83 | 7 | 191 | 13 | 3 |
| Rebuttal/Reservation | 23 | 1 | 0 | 13 | 0 | 0 |
| General Content | 50 | 80 | 18 | 86 | 13 | 13 |
| Evidence | 30 | 7 | 4 | 25 | 1 | 0 |
| *Surface* | 0 | 0 | 466 | 0 | 0 | 422 |
| Word Usage/Clarity | 0 | 0 | 362 | 0 | 0 | 357 |
| Conventions/Grammar/Spelling | 0 | 0 | 75 | 0 | 0 | 52 |
| Organization | 0 | 0 | 29 | 0 | 0 | 13 |

Table 3: Number of revisions, by fine-grain revision purposes and edit types (add, delete, modify).

determine correlation between quantitative measures, we conduct Pearson and Spearman correlation tests.

### 4.1.3 Results and Discussions

**Testing for H1** Comparing Group A and Group B participants, we observe some differences. First, we detect that Group A agrees with the statement "The system helps me to recognize the weakness of my essay" more so than Group B (Group A has a mean ratings of 3.97 ("Agree") while Group B's is 3.17 ("Neutral"), $p < .003$). Second, in Group A, there is a trending positive correlation between the number of revisions from Draft2 to Draft3 and the ratings for the statement "The system encourages me to make more revisions than I usually make" ($\rho$=.33 and $p < .07$); whereas there is no such correlation for Group B. Additional information about revision purposes may elicit a stronger self-reflection response in Group A participants. In contrast, in Group B, there is a significant negative correlation between the number of revisions from Draft1 to Draft2 and ratings for the statement "it is convenient to view my previous revisions with the system" ($\rho$=-.36 and $p < .05$). This suggests that the character-based interface is ineffective when participants have to reflect on many changes.

On the other hand, when comparing the number of revisions made by Group A and Group B on Rev23 (controlling for their Rev12 numbers), we did not find a significant difference.

With no main effect for interface group we cannot verify that **H1** is true; possibly a larger pool of participants is needed; or possibly the assignment is not extensive enough (in length and in the number of drafts).

**Testing for H2** Focusing on the 30 participants from Group A, we check the impact of the feed-back Rev12 on how they subsequently revise (Rev23). We counted the number of times in which the participant disagrees with the revision purpose assigned by the annotator in Rev12. Of those, we then count the number of times the corresponding sentences were further revised (i.e., in Rev23). Of the 53 sentences where the participants disagreed with the annotator, 45 were further revised in the third draft. The ratio is 0.849, much higher than the overall ratio of general Rev12 revisions being further revised in Rev23 (161/394 = 0.408) and the ratio of the agreed Rev12 revisions being revised in Rev23 (67/341 = 0.196). In further analysis, a Pearson correlation test is conducted to check the correlation between the number of Rev23 and the number of disagreements for different types of agreement/disagreements, controlling for the number of Rev12. We find a negative correlation between Rev23 and the number of cases in which the revisions annotated as *Content* are verified by the participants; we also find a positive correlation between Rev23 and the number of cases in which the revisions annotated as *Surface* are intended to be *Content* revisions by the participants. Both findings are consistent with **H2**, which suggests that participants will revise further if they perceive that their intended revisions were not recognized.

**Testing for H3** We observe that native and L2 speakers exhibit different behaviors. First, we detect a significant difference in the number of Content and Surface revisions made by L2 and native speakers ($p < .02$ and $p < .03$). More specifically, native speakers tend to make more *Content* changes while the L2 speakers are likely to make more *Surface* changes. Second, there is also a significant interaction effect among two factors of Group and users' native language ($p < .021$) on their ratings for the statement "the system helps

me to recognize the weakness of my essay". Third, we observe a significant positive correlation in the native group between the number of content revisions in Rev23 and the ratings of the statement "the system encourages me to make more revisions than I usually make" ($\rho$=.4 and $p < .009$). This suggests that giving feedback (from either interface) encourages native speakers to make more content revisions. Finally, in the L2 group, there is a significant negative correlation between the number of surface revisions in Rev12 and the ratings for the statement "the system helps me to recognize the weakness of my essay" ($\rho$=-.57 and $p < .008$). This shows that giving feedback to L2 speakers is less helpful when they make more surface revisions. These results are consistent with **H3**.

**Summary**  Experimental findings over the three hypotheses suggest that feedback on revisions do impact how students review and rewrite their drafts. However, there are many factors at play, including the interface design and the students' linguistic backgrounds.

## 4.2 Application: Automatic Revision Identification

Another application of the corpus is to serve as the gold standard for training and evaluation of an automatic revision analysis system. One subtask of such a system is to predict the intention of the revision purpose. This task was previously investigated by Zhang and Litman (2015). They developed and reported the performance of a binary classifier for each individual revision category using features from the argument mining and discourse analysis researches. The availability of our corpus makes it possible for researchers to replicate their findings and conduct further studies.

### 4.2.1 Hypotheses

In this paper, we repeat the experiment of Zhang and Litman (2015) under different settings to investigate three new hypotheses that can now be investigated given the features of our corpus:

**H4**.  The method used in Zhang and Litman (2015) for high school writings is also useful for the writings of college students.

**H5**.  The same revision classification method works differently for first revision attempts and second revision attempts.

**H6**.  The revision classification model trained

on L2 essays has a different preference from the model trained on native essays.

### 4.2.2 Methodology

We followed the work in Zhang and Litman (2015), where the unigram feature was used as the baseline and the SVM classifier was used as the classifier.  Besides the unigram feature, three groups of features that were used in argument mining and discourse analysis researches were extracted (*Location*, *Textual* and *Language*) (Burstein et al., 2001; Falakmasir et al., 2014). Unweighted average F-score for each category is reported.

For **H4**. 10-fold (participant) cross-validation is conducted on all the essays in the corpus. We compared the results using unigram features and the results using all the features. If **H4** is true, we should expect a similar improvement over the unigram baseline using our corpus.

For **H5**. 10-fold cross-validation was conducted for the revisions from Draft1 to Draft2 and revisions from Draft2 to Draft3 separately. We compared the improvement ratio brought by the advanced features over the unigram baseline.

For **H6**. In this experiment we trained two classifiers separately with L2 and native essays with all the features.  20 native participants were first randomly selected as the test data.  Afterwards classifiers were trained separately using the 20 L2 participants' essays and the remaining 20 native participants' essays. We would expect that the performance of the two trained classifiers is different on the same test data.

### 4.2.3 Results and Discussion

The first two rows of Table 4 support **H4**. We observe that the method used in Zhang and Litman (2015) significantly improves performance (compared to a unigram baseline) for half of the classification tasks, which is similar to Zhang and Litman's results on high school (primarily L1) writing.  In our corpus, performance on *Claim*, *Evidence*, *Rebuttal* and *Organization* was not significantly better than the baseline, possibly due to the limited number of training samples for these categories (Table 3).

For **H5**, the four rows in the middle of Table 4 show the difference of the cross-validation results on first attempt revisions and second attempt revisions. The earlier results using all the revisions, versus now just using only Rev12 or Rev23 re-

| Experiments | Text-based | | | | | Surface | | |
|---|---|---|---|---|---|---|---|---|
| | Claim | Warrant | General | Evidence | Rebuttal | Org. | Word | Conv |
| 10fold + All Revs + Unigram | 0.49 | 0.58 | 0.48 | 0.49 | 0.49 | 0.49 | 0.73 | 0.49 |
| 10fold + All Revs + All features | 0.49 | 0.77∗ | 0.55∗ | 0.50 | 0.49 | 0.49 | 0.86∗ | 0.62∗ |
| 10fold + Rev12 + Unigram | 0.50 | 0.58 | 0.47 | 0.50 | 0.50 | 0.50 | 0.57 | 0.62 |
| 10fold + Rev12 + All features | 0.50 | 0.77∗ | 0.56∗ | 0.50 | 0.50 | 0.50 | 0.72∗ | 0.72∗ |
| 10fold + Rev23 + Unigram | 0.50 | 0.46 | 0.53 | 0.49 | 0.50 | 0.50 | 0.58 | 0.46 |
| 10fold + Rev23 + All features | 0.50 | 0.60∗ | 0.65∗ | 0.49 | 0.50 | 0.50 | 0.78∗ | 0.50 |
| 20 L2 (train) + 20 Native (test) | 0.50 | 0.72 | 0.48 | 0.49 | 0.50 | 0.50 | 0.83 | **0.63** |
| 20 Native (train) + 20 Native (test) | 0.50 | **0.76** | **0.52** | 0.49 | 0.50 | 0.50 | **0.89** | 0.54 |

Table 4: Average unweighted F-score for each binary classification task; The first 6 rows shows the average value of 10-fold cross-validation. ∗ indicates significantly better than unigram baseline ($p <$ .05); The last 2 rows shows the F-value for training on L2/Native data and testing on Native data. **Bold** indicates larger than the number in the other row.

visions are similar, which provides little support for **H5**. With one exception, the features proposed in (Zhang and Litman, 2015) could again significantly improve the performance over the unigram baseline, for the same set of categories as when using all the revisions. However, for the *Conventions/Grammar/Spelling* category, we did not observe a significant improvement for revisions from Draft2 and Draft3. A possible explanation is that there is a bigger difference in the writers' rewriting behavior from Draft2 to Draft3, which increases the difficulty of prediction.

The last two rows of Table 4 support **H6**. Interestingly, we observe a better performance on *Warrant*, *General* and *Word Usage/Clarity* with a classifier trained and tested using native essays. Perhaps essays of native speakers are more similar to each other when revised along these dimensions. For *Conventions/Grammar/Spelling*, in contrast, the classifier trained on L2 data yields a better performance on the same native test set. This may be because the L2 revisions usually include more spelling/grammar corrections.

## 5 Conclusion and Future Works

In this paper we present a new corpus for writing comparison research. Currently the corpus focuses on revision analysis over essay revisions made by both native and L2 college students. In addition to three drafts of essays, we have compared and analyzed the drafts to align semantically identical sentences and assigned revision purposes for each aligned revision sentence pairs. We have also conducted two separate studies to demonstrate the application of the corpus for revision behavior analysis and for automatic revision purpose classification.

We plan to further augment the corpus to advance research on revision analysis in the future. Some potential augmentations include: more fine-grained revision categories, other properties of the revisions, such as statement strength (Tan and Lee, 2014), sub-sentential scopes, and evaluative measures on the revisions.

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
