# Peer review of "A Corpus of Annotated Revisions for Studying Argumentative Writing"

_ACL 2017 — decision unknown_

[Official Review · Reviewer 1 · rating 2 · confidence 4]
soundness 3 · originality 4 · clarity 5 · impact 3 · substance 3 · appropriateness 3 · meaningful comparison 3 · presentation format Poster

This paper presents a corpus of annotated essay revisions. 

It includes two examples of application for the corpus:

1) Student Revision Behavior Analysis and 2) Automatic Revision Identification

The latter is essentially a text classification task using an SVM classifier
and a variety of features. The authors state that the corpus will be freely
available for research purposes.

The paper is well-written and clear. A detailed annotation scheme was used by
two
annotators to annotate the corpus which added value to it. I believe the
resource might be interesting to researcher working on writing process research
and related topics. I also liked that you provided two very clear usage
scenarios for the corpus. 

I have two major criticisms. The first could be easily corrected in case the
paper is accepted, but the second requires more work.

1) There are no statistics about the corpus in this paper. This is absolutely
paramount. When you describe a corpus, there are some information that should
be there. 

I am talking about number of documents (I assume the corpus has 180 documents
(60 essays x 3 drafts), is that correct?), number of tokens (around 400 words
each essay?), number of sentences, etc. 

I assume we are talking about 60 unique essays x 400 words, so about 24,000
words in total. Is that correct? If we take the 3 drafts we end up with about
72,000 words but probably with substantial overlap between drafts.

A table with this information should be included in the paper.

2) If the aforementioned figures are correct, we are talking about a very small
corpus. I understand the difficulty of producing hand-annotated data, and I
think this is one of the strengths of your work, but I am not sure about how
helpful this resource is for the NLP community as a whole. Perhaps such a
resource would be better presented in a specialised workshop such as BEA or a
specialised conference on language resources like LREC instead of a general NLP
conference like ACL.

You mentioned in the last paragraph that you would like to augment the corpus
with more annotation. Are you also willing to include more essays?

Comments/Minor:

- As you have essays by native and non-native speakers, one further potential
application of this corpus is native language identification (NLI).

- p. 7: "where the unigram feature was used as the baseline" - "word unigram".
Be more specific.

- p. 7: "and the SVM classifier was used as the classifier." - redundant.